# Is it possible to estimate aerosol optical depth from historic colour paintings?

Christian von Savigny[1], Anna Lange[1], Anne Hemkendreis[2], Christoph G. Hoffmann[1], and Alexei Rozanov[3]

[1]Institute of Physics, University of Greifswald, Felix-Hausdorff-Str. 6, 17489 Greifswald, Germany
[2]Institute of Art History, Albert-Ludwigs-University of Freiburg, Platz der Universität 3, 79085 Freiburg, Germany
[3]Institute of Environmental Physics, University of Bremen, Otto-Hahn-Allee 1, 28359 Bremen, Germany

**Correspondence:** Christian von Savigny (csavigny@physik.uni-greifswald.de)

**Abstract.**

The idea of estimating stratospheric aerosol optical thickness from the twilight colours in historic paintings – particularly under conditions of volcanically enhanced stratospheric aerosol loading – is very tantalising, because it would provide information on the stratospheric aerosol loading over a period of several centuries. This idea has in fact been applied in a few studies in order to provide quantitative estimates of the aerosol optical depth after some of the major volcanic eruptions that occurred during the past 500 years. In this study we critically review this approach and come to the conclusion that the uncertainties of the estimated aerosol optical depths are so large that the values have to be considered questionable. We show that several auxiliary parameters – which are typically poorly known for historic eruptions – can have a similar effect on the red-green colour ratio as a change in optical depth typically associated with eruptions such as, e.g. Tambora in 1815 or Krakatoa in 1883. Among the effects considered here, uncertainties in the aerosol particle size distribution have the largest impact on the colour ratios and hence the aerosol optical depth estimate. For solar zenith angles exceeding 80°, uncertainties in the stratospheric ozone amount can also have a significant impact on the colour ratios. In addition, for solar zenith angles exceeding 90° the colour ratios exhibit a dramatic dependence on solar zenith angle, rendering the estimation of aerosol optical depth highly challenging. A quantitative determination of the aerosol optical depth may be possible for individual paintings, for which all relevant parameters are sufficiently well constrained in order to reduce the related errors.

## 1   Introduction

Volcanic eruptions constitute one of the largest uncertainties for natural climate variability on time scales of a few years up to a decade (e.g. Robock, 2000; von Savigny et al., 2020). Prominent examples of volcanic eruptions with significant climatic effects are the 1815 eruption of Tambora, which led to the year without summer in 1816 (Raible et al., 2016), or the 1991 eruption of Mt. Pinatubo. Volcanic eruptions provide an important opportunity to improve the scientific understanding of the climate system's response to perturbations. Several sources of information are available for investigating past volcanic activity, e.g. ice cores (e.g. Zielinski et al., 1994), tree-rings (e.g. Esper et al., 2017) or historic reports (e.g. Bauch, 2017). Volcanic eruptions can also lead to significant changes in the colour of the sky, as well as to unusual optical phenomena such as blue or

green suns (e.g. Symons et al., 1888; Horvath et al., 1994; Wullenweber et al., 2021) and Bishop's rings (Symons et al., 1888). For detailed accounts of unusual optical phenomena associated with volcanic eruptions we refer to the classical works by Kiessling (1888) and Symons et al. (1888). Some studies employed the sky colours in historic paintings to detect evidence for volcanic eruptions and to infer quantitative information on the aerosol optical depth at the time, when the painting was painted (e.g. Zerefos et al., 2007, 2014). The basic idea is to extract the red-green colour ratios of selected areas of the evening sky in photographs of historic paintings and relate these ratios to ratios simulated with a radiative transfer model in order to derive quantitative information on the aerosol optical depth. The earlier studies mentioned above focused on the evening sky before or after sunset, because anomalous stratospheric aerosol layers affect the colours of the evening sky. Note that similar effects will also occur during sunrise. The use of red-green colour ratios is motivated by the fact that a variable stratospheric aerosol loading will lead to changes in the spectral distribution of the scattered solar radiation perceived by a ground-based observer. These changes can be caused by the changing relative importance of Rayleigh and Mie-scattering and also by changes in the particle size distribution of aerosols.

In the present study we scrutinise the robustness of the very interesting approach of aerosol optical depth estimations based on colour ratios extracted from photographs of historical paintings and investigate the effect of several additional parameters or processes that also affect the colour ratios, but that were not considered properly in earlier studies. We also briefly discuss some additional general aspects, e.g. whether painters can be expected to reproduce realistic colours in the works, or how the colours of a painting may change over time. Note that our initial motivation was to support the approach by providing a better theoretical foundation.

The paper is structured as follows. In section 2 we introduce the radiative transfer model SCIATRAN used in this study to simulate colour ratios. In addition, we briefly introduce the concept of the CIE (International Commission on Illumination) XYZ-tristimulus values that allows for a more realistic simulation of colours perceived by humans than the use of two discrete wavelengths. Section 3 presents simulation results on the dependence of colour ratios and X/Y tristimulus value ratios on several relevant parameters, e.g. the particle size distribution (PSD) of the volcanic aerosol layer, the amount of stratospheric ozone or surface albedo. Section 3 also compares our results to previous studies on this topic. A general discussion of the results is provided in section 4 and conclusions are given at the end.

## 2 Methodology

We first describe the Mie simulations and the radiative transfer model employed, followed by a summary of the colour modelling approach used in this study.

### 2.1 Radiative transfer simulation: Mie scattering calculations and SCIATRAN

The radiative transfer simulations presented in this work were carried out with version 4.1.3 of the SCIATRAN radiative transfer model (Rozanov et al., 2014). SCIATRAN is a highly versatile model initially developed for the analysis of remote sensing measurements in the optical spectral range. In the present work, SCIATRAN was run in the approximate spherical mode, where

the contribution of single scattering is calculated in a fully spherical geometry, while an approximation is employed to account for the multiple scattering contribution (Rozanov et al., 2000). As discussed by Lange et al. (2022) this approximation is fully sufficient to analyze the colors of the horizon area during twilight. Both the sun-ward near-horizon radiances and the diffuse downward fluxes as analyzed by Zerefos et al. (2007) were modeled with SCIATRAN. The SCIATRAN input parameters relevant to this study are given in Table 1. Figure 1 provides an illustration of the observation geometry and the relevant angles. Note that the range of solar azimuth angles considered implies that the observer is looking in sun-ward direction. The volcanic stratospheric aerosol layer considered here has a Gaussian shape of the particle number density profile with a peak altitude of $z_{peak} = 20\,$km and a FWHM (Full Width at Half Maximum) of 3 km. The peak altitude and the FWHM were also varied, causing only minor effects on the results. For this reason, only results for $z_{peak} = 20\,$km and FWHM = 3 km are presented in this study. The troposphere was assumed to be free of aerosols. We did test the effect of tropospheric aerosols with an optical depth of 10% of the total aerosol optical depth – i.e. $\tau_{tropo} = 0.03$ for a $\tau_{total} = 0.3$ – and the effect on the colour ratios was less than 0.5% for SZA < 90°. For SZA > 90° differences of up to about 10% can occur. Note that a tropospheric aerosol optical depth of $\tau_{tropo} = 0.03$ is lower than the current globally averaged value of $\tau_{tropo} = 0.12 \pm 0.04$ (Ramanathan et al., 2001), but the tropospheric aerosol optical depth is highly variable and values of $\tau = 0.03$ are observed with the Aeronet photometer operated at the Institute of Physics of the University of Greifswald (results not shown). We may assume that the aerosol optical depth was even lower in preindustrial times. In addition, the overall conclusions of this study are not affected by the assumption of an aerosol free troposphere. Variable or unknown abundance and characteristics (e.g. composition or vertical distribution) of a tropospheric aerosol component adds even more complexity. More detailed information on the SCIATRAN radiative transfer model is available in Rozanov et al. (2014) and the SCIATRAN User's Guide (IUP, 2021).

We assume the volcanic aerosol layer to consist of sulfate particles and the refractive index was chosen correspondingly, i.e. $n_r = 1.43 - 1 \times 10^{-8} i$ at a wavelength of 550 nm. Note that the spectral dependence of the refractive index is considered in this study. The refractive index data is taken from the OPAC (Optical Properties of Aerosols and Clouds) database implemented in SCIATRAN (IUP, 2021). Further, we assume a mono-modal log-normal particle size distribution:

$$n(r,z) = \frac{N_0(z)}{\sqrt{2\pi} \cdot \ln(S) \cdot r} \cdot exp\left[-\frac{(\ln r - \ln r_m)^2}{2\ln^2(S)}\right] \tag{1}$$

with $N_0$ being the particle number density, $r_m$ the median radius, $r$ the particle radius and $S$ the geometric standard deviation of the distribution. Mie scattering phase functions and scattering cross-sections are determined with the Mie scattering routines included in the SCIATRAN radiative transfer model. The simulated spectra were multiplied with a solar irradiation spectrum based on SORCE measurements (LASP, 2003). In addition, we used standard pressure and temperature as well as atmospheric trace gas profiles for tropical latitudes and the month of June taken from a climatological database based on simulations with the 3-D chemical transport model by Sinnhuber et al. (2003).

Using the SCIATRAN calculations of near-horizon radiance or diffuse downward flux spectra covering the 300 nm to 800 nm spectral range, we determine colour ratios based on (a) discrete wavelengths and (b) ratios of the CIE X and Y tristimulus values – which consider certain wavelength ranges in contrast to the discrete wavelengths in (a) – as described below. The

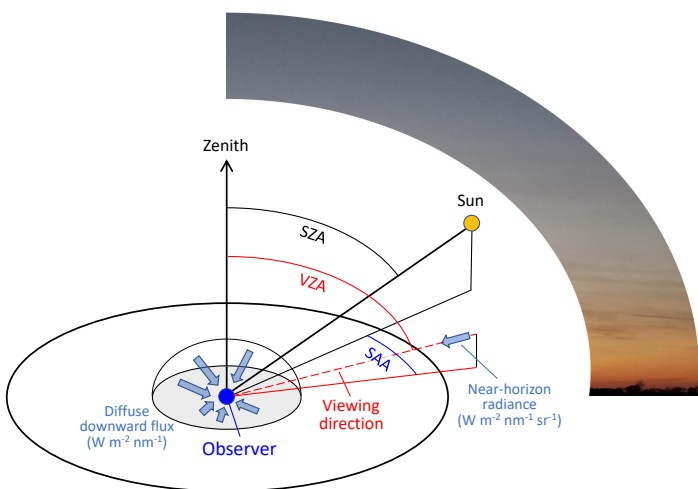

**Figure 1.** Illustration of the observation geometry, the relevant angles (i.e. solar zenith angle (SZA), viewing zenith angle (VZA), solar azimuth angle (SAA)), the diffuse downward flux and the near horizon radiance. Note that the diffuse downward flux cannot be represented perfectly in a 2-D plot (please see text for more information).

| Parameter | Abbreviation | Baseline value | Perturbed values |
|---|---|---|---|
| Viewing zenith angle | VZA | $85°$ | $70°, 75°, 80°, 90°$ |
| Solar azimuth angle* | SAA | $10°$ | $20°, 30°, 40°$ |
| Stratospheric aerosol optical depth | AOD | N/A | 0.0, 0.05, 0.1, 0.2, 0.3 |
| Median radius$^\dagger$ | $r_{med}$ | 250 nm | 150 nm, 350 nm, 450 nm |
| Geometric width$^\dagger$ | S | 1.5 | N/A |
| Total ozone column | TOC | 300 DU | 200 DU, 400 DU |
| Surface albedo | $A_s$ | 0.3 | 0.1, 0.5, 0.7, 0.9 |

**Table 1.** Compilation of the relevant input parameters for the SCIATRAN simulations. Note that the solar zenith angle (SZA) is varied for all case studies in the range from $40° - 96°$ (* the solar azimuth angle corresponds to the azimuthal angular difference between the viewing direction and position of the sun for a ground-based observer, i.e. an SAA of $0°$ means that the observer is looking in the direction of the sun; $^\dagger$ parameters of a mono-modal log-normal particle size distribution (see Eqn. 1)).

tristimulus values are determined by weighting the simulated spectra with the $x(\lambda)$ and $y(\lambda)$ CIE (International Commission on Illumination) colour matching functions, as described in more detail in, e.g. Wullenweber et al. (2021). The CIE colour

matching functions represent the spectral sensitivity functions of the three types of cone cells responsible for human colour vision and are displayed in Fig. 2.

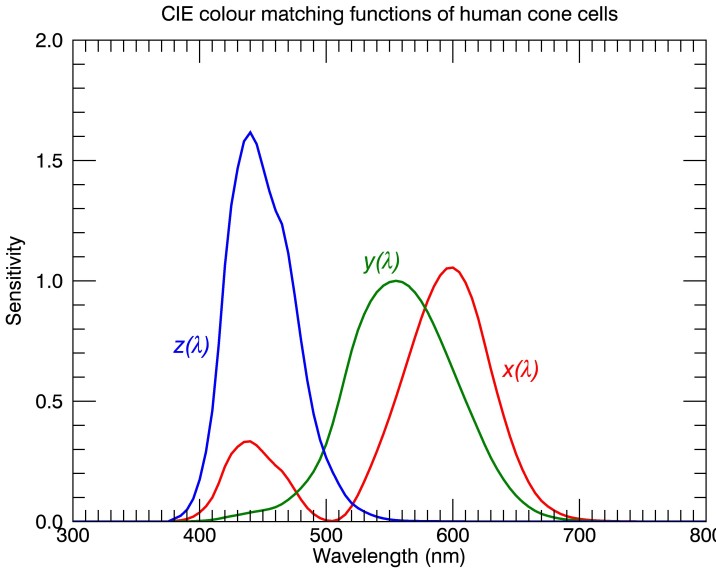

**Figure 2.** CIE colour matching functions of the human eye for the CIE 1931 2°-standard observer, after Judd (1951) and Vos (1978) (http://cvrl.ioo.ucl.ac.uk/cmfs.htm, last checked: October 4, 2022).

## 3 Results

We start in section 3.1 with a brief discussion of the wavelengths used to form the colour ratios based on radiative transfer simulations. In the following sections 3.2 to 3.7 we discuss the effects of different parameters and processes on colour ratios of the evening sky that challenge the estimation of aerosol optical depth from colour ratios extracted from historic paintings.

### 3.1 Choice of wavelengths for colour ratio simulations

Zerefos et al. (2007) performed radiative transfer simulations and determined the colour ratio using the wavelengths of 700 nm and 550 nm. One issue is that they did not calculate the radiance ratio for a near-horizon viewing geometry, but calculated the ratio of the diffuse downward fluxes at the two wavelengths, although the near-horizon radiance would be the correct quantity in this context. Colour ratios of near-horizon radiances and diffuse downward fluxes show quite a different dependence on AOD, as will be shown in section 3.7 below. Radiance corresponds to the number of photons per unit area, per unit time and per unit solid angle for a given direction, whereas the diffuse downward flux corresponds to the radiance integral over all directions of the upper half sphere, weighted by the cosine of the incidence angle, which gives more weight to smaller incidence angles

and almost eliminates contributions from incidence angles close to 90°, corresponding to directions near the horizon. In the current section we focus on the choice of wavelengths. While 550 nm coincides well with the maximum of the $y(\lambda)$ CIE colour matching function – corresponding to the green-sensitive cones – 700 nm is quite far off the maximum of the $x(\lambda)$ colour matching function, representing the red-sensitive cones (see Fig. 2). The maximum of $x(\lambda)$ is at about 600 nm. For this reason, 700 nm appears to be an unsuitable choice in order to relate modelled colour ratios to red-green colour ratios extracted from

pictures of paintings. In addition, at wavelengths of around 700 nm the spectra are affected by an $H_2O$ absorption band that can have a significant impact on the radiance values (see Fig. 3). For this reason, we perform our first tests with a wavelength of 670 nm – rather than 700 nm – which is relatively free of absorption (Fig. 3). Note that we employ 670 nm in order to use a wavelength as close as possible to the wavelength used in Zerefos et al. (2007), but to avoid the $H_2O$ absorption band near 700 nm.

In the following, we will also determine colour ratios based on the X and Y tristimulus values, which better represent the colour sensitivity of the human eye's cone cells. The red-green colour ratio is then represented by the ratio X/Y. Note that the scattering spectra simulated with SCIATRAN could also be transformed to RGB-values – which at first glance appears to be the best possibility to compare simulations with red-green colour ratios determined from pictures. However, the transformation from the XYZ-tristimulus values to RGB-values requires additional assumptions and introduces uncertainties and is therefore

not carried out in this study.

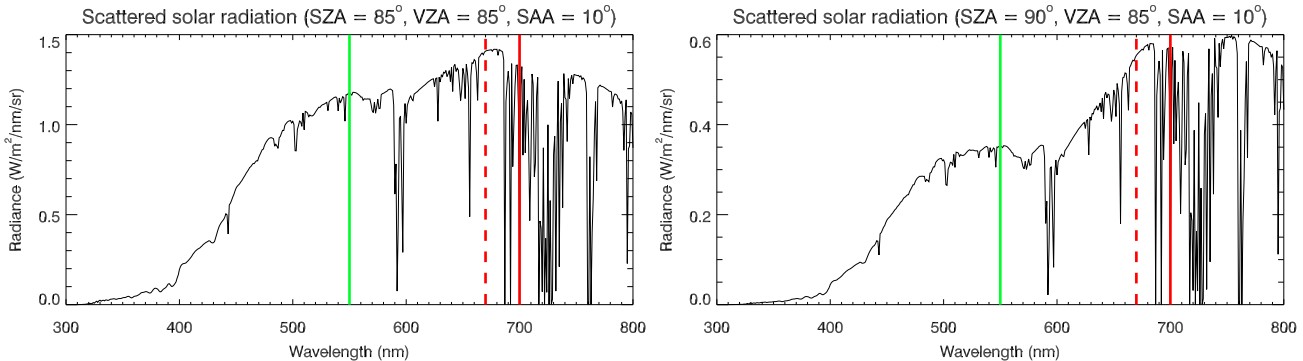

**Figure 3.** Sample spectra of scattered solar radiation for SZAs of 85° (left panel) and 90° (right panel) and the following parameters: VZA = 85°, SAA = 10°, TOC = 300 DU, $A_S$ = 0.3, AOD = 0.1, $r_{med}$ = 250 nm and S = 1.5. The green and red solid lines indicate wavelengths of 550 nm and 700 nm, while the dashed red line corresponds to 670 nm.

## 3.2 Dependence on optical depth

The left panel of Fig. 4 shows the SZA dependence of the 670 nm/550 nm radiance ratio for different values of the stratospheric AOD as grey lines. The AOD values are: 0.0, 0.05, 0.1, 0.2 and 0.3, covering the vast majority of the eruptions of the past

millennium. The grey lines show simulations with constant aerosol size distribution parameters ($r_{med}$ = 250 nm and S = 1.5),
TOC = 300 DU, a surface albedo of 0.3 and viewing angles of VZA = 85° and SAA = 10°. Apparently and as expected, the
670 nm/550 nm radiance ratio is affected by the AOD and increases with increasing AOD value. It is interesting that the change
in radiance ratio per 0.1 AOD step decreases with increasing AOD, consistent with Fig. 4 in Zerefos et al. (2007). For SZAs
exceeding 90° the 670 nm/550 nm radiance ratio increases strongly with SZA and reaches values exceeding 20 at SZA = 98°.
This strong SZA dependence is not observed in Fig. 4 of Zerefos et al. (2007), because their Fig. does not show the ratio of
evening sky radiances, but the ratio of the diffuse downward fluxes (see also section 3.7).

The right panel of Fig. 4 shows the SZA dependence of the ratio of the tristimulus values X/Y, which is a more appropriate
colour ratio to be compared to red-green-ratios extracted from historic paintings, as explained above. Qualitatively, the SZA
and AOD dependence of the X/Y ratio is similar to the 670 nm to 550 nm colour ratio. However, the SZA dependence for SZA
> 90° is not as strong if the ratio of the tristimulus values X and Y is used. Still, also the X/Y-ratio varies rapidly with changing
SZA for SZA > 90°, rendering a quantitative determination of the AOD for SZA > 90° essentially impossible, if the actual
SZA is not well known. Similar colour ratios can be produced with highly different AODs, but for different solar and viewing
angles. Assuming a constant value of SZA = 100° for all paintings with SZA > 90° as in Zerefos et al. (2007) will certainly
not work. Another issue related to the strong SZA-dependence of the colour ratios for SZA > 90° is the fact that evening sky
colours can probably not be captured immediately in an oil painting, but the painting will take a certain time to be finished.
During sunset the illumination conditions change continuously and with them also the colours of the evening sky.

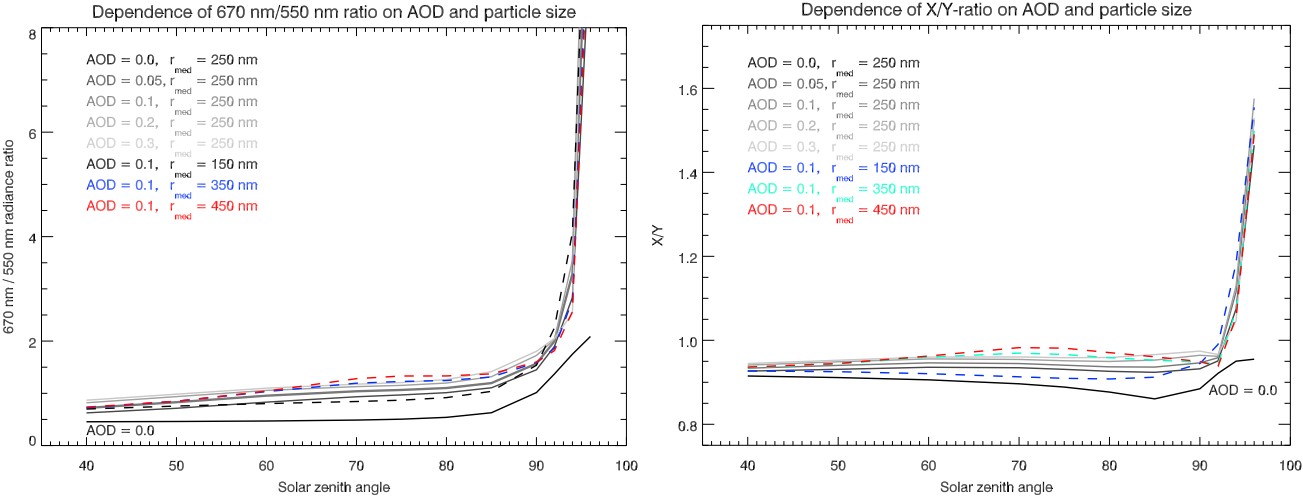

**Figure 4.** Ratio of the scattered solar radiances at 670 nm and 550 nm (left panel) and the X/Y tristimulus value ratio (right panel) as a function of SZA and for different values of the stratospheric AOD and different aerosol size distributions. Note that for SZAs < 90° the radiance ratio increases monotonously with AOD for the AOD range shown here. Other relevant parameters: VZA = 85°, SAA = 10°, TOC = 300 DU, $A_S$ = 0.3, S = 1.5.

### 3.3 Dependence on particle size

The scattering spectrum and also the colour ratios may depend sensitively on the particle size distribution of the stratospheric aerosol. In order to test this sensitivity we performed SCIATRAN simulations for different assumptions on the PSD. In all cases the PSD was assumed to be mono-modal log-normal and the geometric width was assumed to be S = 1.5. For the median radius the following values were chosen: 150 nm, 250 nm, 350 nm and 450 nm. Note that the actual PSD of stratospheric aerosols can be highly variable and may not always follow a mono-modal log-normal distribution (e.g. Deshler, 2008), but in many cases this assumption describes the actual PSD well. For background conditions typical values of the median radius are in the range 50 – 150 nm and for volcanically disturbed conditions, median radii exceeding 400 nm were observed, e.g. after the eruption of Mt. Pinatubo in 1991 (Bingen et al., 2004). It is important to mention that the variation of the particle size distribution of stratospheric sulfate aerosols after volcanic eruptions is not fully understood (Robock, 2000) and topic of current research. Here we attempted to make assumptions on the particle size parameters that are consistent with the current scientific understanding. We also note that systematic differences between particle size estimates can occur if different observation geometries are used (von Savigny and Hoffmann, 2020). In a recent study, Thomason et al. (2021) showed first evidence that the mean aerosol particle size decreases after many moderate volcanic eruptions. Since the eruption of Mt. Pinatubo in 1991 led to a well observed increase in the particle size of stratospheric aerosols (e.g. Bingen et al., 2004; Deshler, 2008), it appears justified to assume an increase in particle size for strong eruptions associated with AODs exceeding $\approx 0.1$, which are relevant for the present study. Note that an accurate knowledge of the PSD and its variation in the aftermath of volcanic eruptions is not crucial for the current study – we only want to test the sensitivity of the colour ratios to the PSD within a plausible range of size parameters.

The left panel of Fig. 4 shows the SZA-dependence of the 670 nm/550 nm radiance ratio for different median radii and a constant AOD of 0.1 as coloured curves. The other parameters are as listed in the Figure caption. Apparently, the effect of the particle size parameters on the colour ratios is quite large in comparison to the variations introduced by different AODs – particularly in the SZA range between 70° and 90° for which volcanic effects on the colour of the evening sky are typically observed. This implies that a determination of the AOD from colour ratios may be affected by large systematic errors if the actual PSD is not known. We note that Zerefos et al. (2007) do not state which PSD has been assumed for their radiative transfer simulations.

The right panel of Fig. 4 shows a similar plot but for the X/Y tristimulus value ratio. Also here the aerosol particle size distribution has a significant effect on the colour ratio, rendering a quantitative determination of the AOD very challenging, if no information of the aerosol PSD is available.

### 3.4 Dependence on ozone column

Since the overall shape of the scattering spectra can also be significantly affected by the amount of ozone in the atmosphere, we tested the impact of ozone by changing the total ozone column (TOC). For all simulations presented so far a TOC of 300 DU (Dobson Units) was assumed. Fig. 5 shows the resulting SZA-dependence of the red-green colour ratio for TOCs of 200 DU,

300 DU and 400 DU. These values were chosen, because for the majority of locations and seasons the TOC falls within the range between 200 DU and 400 DU. Exceptions would be, e.g. ozone hole conditions with TOC values as low as 100 DU or Arctic spring conditions at northern mid-latitudes with TOC values of up to about 500 DU. Typical values for northern mid-latitudes are between 300 DU and 400 DU. Note that volcanic eruptions may also affect the stratospheric ozone layer. During times of anthropogenically enhanced stratospheric halogen loadings, volcanic eruptions may lead to a decrease in TOC, as clearly observed after the eruptions of El Chichon (1982) and Pinatubo (1991) (e.g. Langematz, 2019). Both eruptions led to a decrease in globally averaged TOC of a few percent. Note that this change is significantly smaller than typical seasonal variations at, e.g. northern mid-latitudes. Before about 1950, however, eruptions may rather have led to a TOC increase (Tie and Brasseur, 1995).

Overall, the effect is significantly smaller than for the variable size parameters, but for SZAs between 80° and 90° an error in the assumed TOC may lead to non-negligible errors in the estimated AOD.

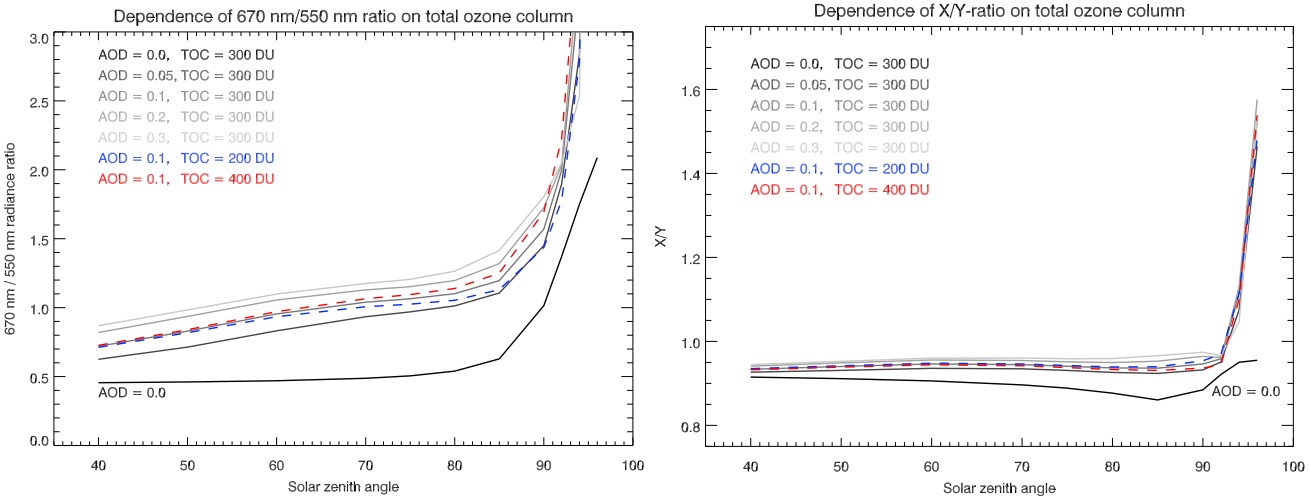

**Figure 5.** Ratio of the scattered solar radiances at 670 nm and 550 nm (left panel) and the X/Y tristimulus value ratio (right panel) as a function of SZA and for different values of the stratospheric AOD and TOC. Other relevant parameters: VZA = 85°, SAA = 10°, $A_S$ = 0.3, $r_{med}$ = 250 nm, S = 1.5.

## 3.5 Dependence on surface albedo

The colour ratios may also depend on the surface albedo and this impact was tested as well. Figure 6 shows a plot similar to Fig. 4 including simulation results for different values of the surface albedo. The albedo effect is relatively small, particularly for large SZAs and should only be a minor issue for the estimation of AOD from the colour ratios, also given the fact that the

surface albedo of large fractions of the Earth is lower than about 0.3. Tropospheric clouds and tropospheric aerosols will also
play a role, but they are not considered in this study.

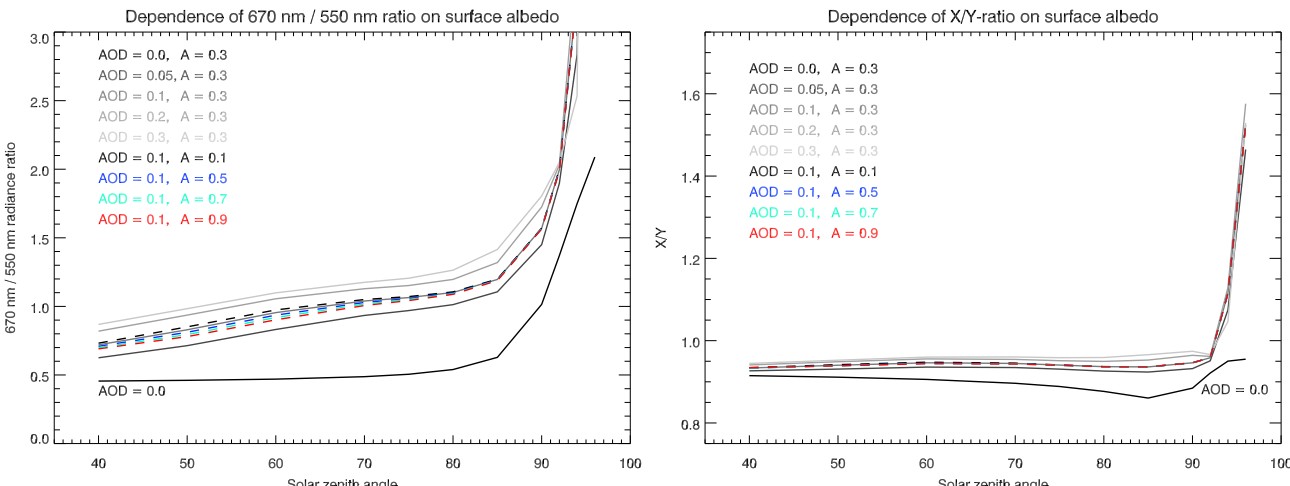

**Figure 6.** Ratio of the scattered solar radiances at 670 nm and 550 nm (left panel) and the X/Y tristimulus value ratio (right panel) as a function of SZA and for different values of the stratospheric AOD and different surface albedos. Other relevant parameters: VZA = 85°, SAA = 10°, TOC = 300 DU, $r_{med}$ = 250 nm, S = 1.5.

## 3.6 Dependence on solar azimuth angle and viewing zenith angle

The colour ratios will certainly also depend on the specific region of the sky viewed. In order to test this effect, we performed simulations for different values of the solar azimuth angle (SAA) and the viewing zenith angle (VZA) (See Fig. 1). As Fig. 7 shows, the results indicate a moderate dependence of the colour ratios on the SAA, but for viewing directions close to the horizon (VZA = 90°) a strong dependence of the colour ratios on the VZA is observed. The results imply that the actual range of viewing angles needs to be carefully considered – and the SZA needs to be precisely known, as mentioned above – when attempting to extract information on the AOD from colour ratios of the evening sky. Otherwise large systematic errors will occur. It is not explicitly mentioned in Zerefos et al. (2007), whether a specific range of viewing angles or a specific solid angle was extracted from photographs of the historic paintings. If the atmosphere close to the horizon is viewed shortly after sunset, relatively large colour ratios occur even for an aerosol-free atmosphere. This agrees with the general notion that colourful red sunsets can also occur in times without recent volcanic activity.

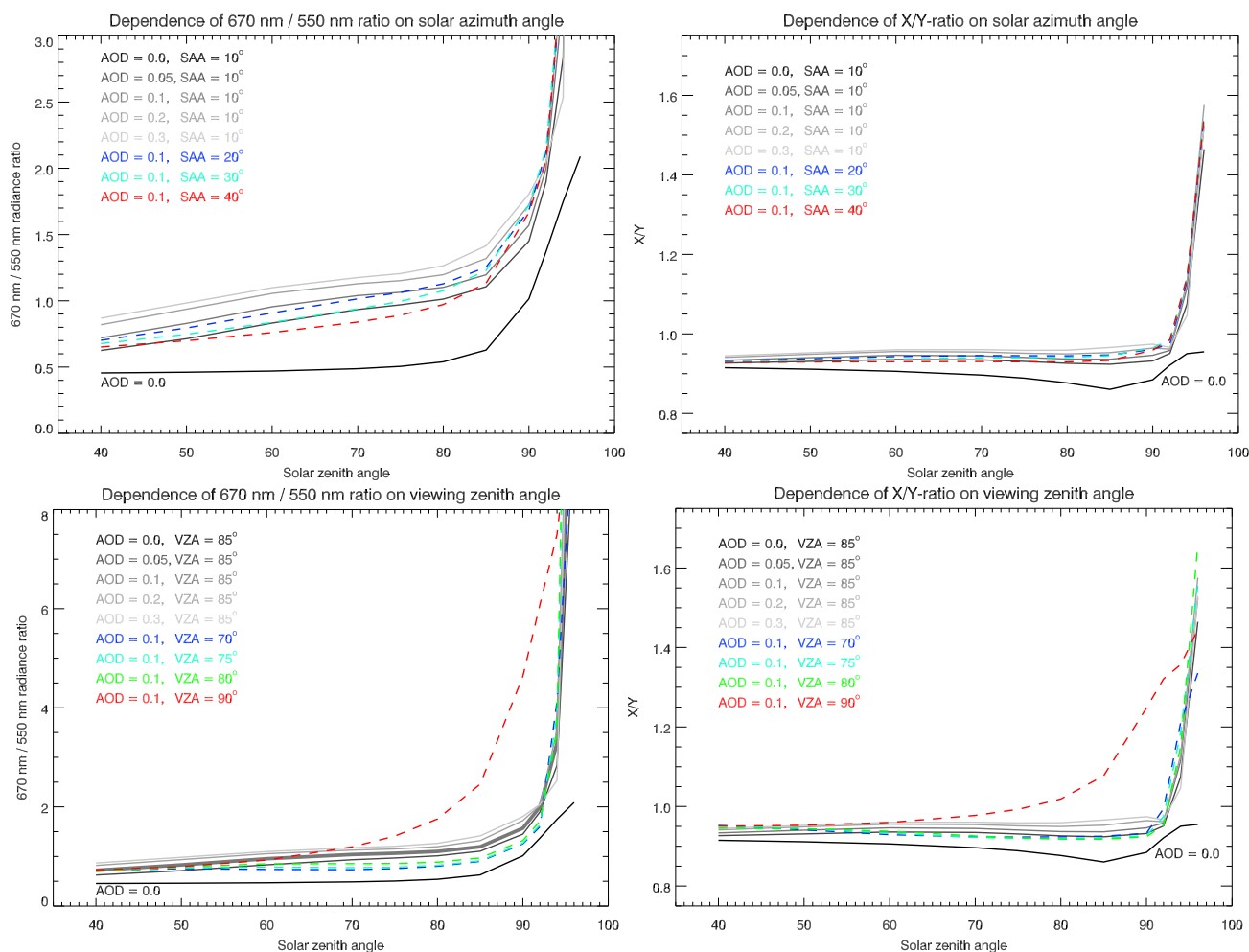

**Figure 7.** Ratio of the scattered solar radiances at 670 nm and 550 nm (left column) and the X/Y tristimulus value ratio (right column) as a function of SZA and for different solar azimuth angles (SAA; upper row), different viewing zenith angles (VZA; lower row) and different values of the stratospheric AOD. Other relevant parameters: TOC = 300 DU, $A_S = 0.3$, $r_{med} = 250$ nm, S = 1.5, VZA = 85° for the upper row and SAA = 10° for the lower row. Note that the ordinate ranges differ between the different panels of this Figure.

### 3.7 Relationship between diffuse downward flux and evening sky radiance

Next, we investigated the relationship between the diffuse downward flux and the evening sky radiance in order to test, whether the diffuse downward flux can be used to infer information on the aerosol optical depth, as in Zerefos et al. (2007). The left
panel of Fig. 8 shows the SZA-dependence of the 670 nm/550 nm colour ratios for both the diffuse downward flux (solid lines) and the near-horizon radiance (dashed lines). The ratios are shown for different values of the AOD, indicated by different colours. Apparently, the radiance ratios exhibit a stronger SZA-dependence than the diffuse flux ratios. Note that the diffuse

flux colour ratios exhibit a similar dependence on aerosol loading and SZA as in Fig. 3 in Zerefos et al. (2007), although the values do not match exactly. These differences could be related to the different assumptions made in Zerefos et al. (2007), e.g.

the exact value of the AOD or the assumed aerosol particle size distribution. Note that the radiance ratios (as compared to the diffuse flux ratios) in the left panel of Fig. 8 are in better overall agreement with the colour ratios derived from historic paintings depicted in Fig. 3 of Zerefos et al. (2007). The results shown in the left panel of Fig. 8 already indicate that the diffuse flux and radiance ratios exhibit quite a different behaviour – as might be expected – which leads to non-trivial systematic errors if AOD is estimated from the diffuse flux ratio rather than the near-horizon radiance ratios. To illustrate the relationship between

radiance and diffuse flux colour ratios further, the right panel of Fig. 8 shows the ratio of the radiance colour ratio and the diffuse flux colour ratio – i.e. the ratio of the dashed and solid lines in the left panel – as a function of SZA. Apparently, this ratio shows a complex dependence on SZA and AOD. The assumption of a constant scaling factor between the diffuse flux ratio and the radiance ratio can lead to large errors in the estimated AOD, if scaled simulations of the diffuse flux ratio are used to estimate AOD from the colour ratios in historic paintings, as in Zerefos et al. (2007).

We conclude that using the diffuse downward flux rather than the evening sky radiance adds additional issues that unnecessarily exacerbate the estimation of aerosol optical depth.

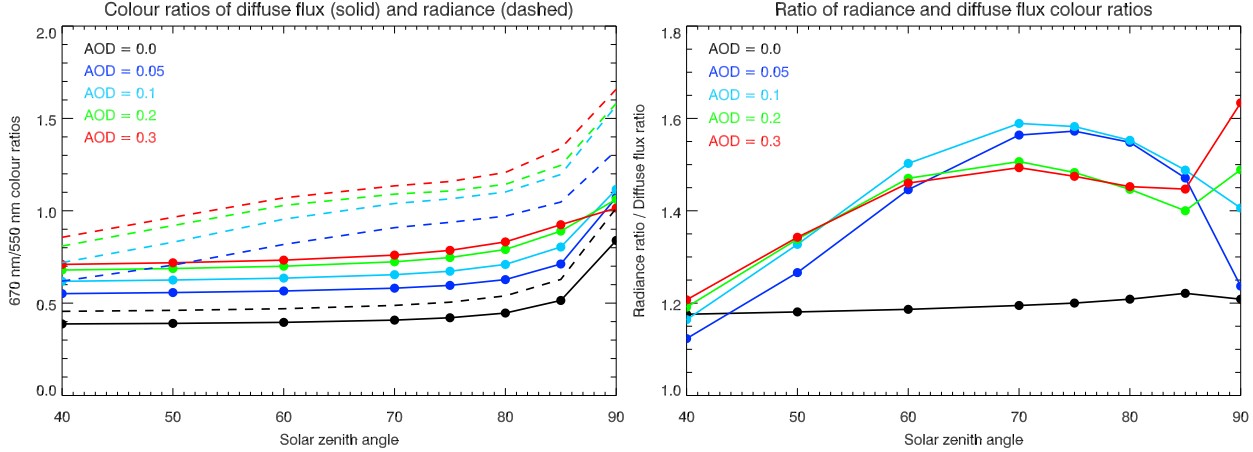

**Figure 8.** Left panel: SZA dependence of the 670 nm/550 nm diffuse downward flux ratio (solid lines) and radiance ratio (dashed lines) for different AOD values. Right panel: SZA dependence of the ratio between the radiance and diffuse flux ratios shown in the left panel, i.e. the ratio of the dashed and solid lines. The following parameters apply to both the diffuse flux and the radiance simulations: TOC = 300 DU, $A_S$ = 0.3, $r_{med}$ = 250 nm, S = 1.5 and the observer is located at the Earth's surface. The viewing angles for the radiance simulations were: VZA = 85° and SAA = 10°.

## 4 Discussion

We want to emphasize that we do not doubt that volcanic effects can be identified in historic paintings. Fig. 3 in Zerefos et al. (2007) demonstrates that there is a systematic difference in the red-green colour ratios between paintings in times without major volcanic eruptions and those painted in the aftermath of such eruptions. These differences are expected and are also present in the simulations shown in this study. The main aspect of this work is to show that a quantitative determination of the AOD from colours in paintings may not be possible in a general way, if not all relevant parameters (see above) are accurately known. It should also be mentioned in this context that Zerefos et al. (2014) carried out a comprehensive comparison of the AOD values inferred from historic paintings with data based on independent methods, including the analysis of ice cores as well as historic optical extinction measurements. The temporal correlations between the AOD values inferred from paintings and from the independent techniques are remarkably high (see Table 1 in Zerefos et al. (2007)). However, the averaged AODs inferred from the paintings – only considering years where AODs from both the paintings and the corresponding independent data set are available – are roughly one order of magnitude larger than the values from the other techniques. This also shows that a quantitative determination of AOD from historic paintings using the approach described in Zerefos et al. (2007, 2014) is associated with large errors.

We note that we do not categorically exclude the possibility of inferring AOD from paintings. It may be possible for individual paintings, for which all relevant parameters are sufficiently well constrained in order to reduce the related errors.

It is also relevant in this context that the Royal Society's Krakatoa report (Symons et al., 1888) includes various reports of unusual twilight colours after the 1883 eruption of Krakatoa that differ significantly from the expected enhanced reddish or violet afterglows. In fact, several observers reported a green colouring of the evening sky after sunset (e.g. "a very curious opalescent shining green and slightly greenish-white", "bright green glow near the place where the sun set", "after sunset yellowish-green striae"). One report states (referring to the year 1884): "During the remainder of the year the sunsets were uncommonly free from colour, even promising skies turning grey soon after sunset, and no redness of an ordinary character remaining along the horizon after sunset, except on a few evenings and in a few localities.". These may be exceptions, but they illustrate that volcanic eruptions may affect sunset colour ratios in a non-uniform way.

Apart from the problems described above there are several additional issues that we discuss in the following section in a qualitative way. Section 4.1 deals with the question, whether the colours in the original paintings can be expected to be realistic, using specific examples from the romanticism period. Section 4.2 addresses the issue, how colours may change due to ageing effects.

### 4.1 Historic Paintings as Documents? Caspar David Friedrich's intense Colours and the Beginning of Climate-Awareness

The study of Zerefos et al. (2007) assumes that paintings of the Romantic period contain precise environmental information. Especially historical paintings created, e.g. in the three years following the 1815 Tambora volcanic eruption are interpreted as visual archives of historical conditions of the atmosphere. Although there was in fact an increasing interest in meteorological

phenomena among artists of the early 19th century – for example John Constable, J. M. W. Turner and Johann Christian Dahl (Gould, 2021; Ogée, 2022) – two objections speak against the interpretation of Romantic paintings as simple data storage. First, the religious meaning of landscape paintings concerning their iconographical as well as philosophical background should not be ignored. Secondly, the differences between natural events and their mediation through art need to be taken into account.

The subsequent analysis focuses on Caspar David Friedrich and is distinguished by the fact that the city view of Greifswald could be examined on site with regard to its visual accuracy. In this context, it is important to note that visual representations do not speak out but speak to, i.e. paintings do not preserve the condition of nature but open up a space for interpretation and contemplation. Among other things, art questions our conditions of perception and our relationship to the world. Thus, although Friedrich's romantic paintings like "Greifswald in Moonshine" (1817) may be based on the experience of a particular natural event, the assumption of its documentary character drastically reduces its complexity of meaning. Realistic representations are often misinterpreted as providing an access to a historical reality.

However, Friedrich's landscapes – including the painting "Greifswald in Moonshine" – are so-called composite landscapes, based on different locations or sketches of vegetative and geological forms to increase the emotional impact on the viewer (Heck, 2015). The landscapes were not painted in nature but inside of Friedrich's studio and took a certain time to be finished. As Kilian Heck argues, a new "dialogue between viewer and image" takes place in front of Friedrich's landscapes, which leads to a conscious examination of one's relationship to nature and God (Heck, 2013).

As a Lutheran, Friedrich shared a general scepticism towards the sense of sight, which inspired him to create a new form of religious art. As Johannes Grave has pointed out, Friedrich's landscape paintings show the artist's way of thinking about the medial conditionalities of images among negotiations of theological questions (Grave, 2011).

This is not to say that Caspar David Friedrich had no interest in the atmospheric changes brought about by the onset of industrialization or that Romantic art in general is not crucial for scientific knowledge production, quite the contrary. Arts and sciences need to be regarded as two sides of the same coin if we want to gain a deeper understanding of climate change, how it is perceived and interpreted. The relatively recent ecological turn in art history has interpreted Friedrich's intense colours not as merely mirroring the artist's interest in "visual spectacles" of the atmosphere, but as a reflection on the "burgeoning idea(s) about the essence of nature" (Amstutz, 2021). In an era of emerging concepts of "geological time and environmental determinism", geologists like Alexander von Humboldt, for example, valued landscape and their climates as factors influencing human physique and character alike (Amstutz, 2021). In this respect, the glow of Friedrich's colours is classified by environmental art historians as an artistic means of creating "an environment (of reception) that had a psychological as well as a physical effect" on the viewer (Amstutz, 2021). Thus, Friedrich's landscape paintings are not collections of unusual weather phenomena. Instead, they reflect on the way in which weather is seen, physically experienced and interpreted with view to one's own subjectivity. Or to put it differently: Friedrich's intense colour palette may be understood as a reaction to the change of the atmospheric conditions after the volcanic eruption, but it is also part of a philosophical discussion on the moral, cultural and physical influence of nature on humankind (Amstutz and Wedekind, 2021). Therefore, Friedrich's painting "Greifswald in Moonshine" is both, an experiment of perception and a pietistic space for reflection.

## 4.2 How do colours in paintings change over time?

Another potential problem is the colour change in historic paintings over time, which is an extremely complex topic. A detailed treatment is well beyond the scope of the present work and we only provide a brief discussion of some basic aspects. Among art historians and art restorers it is well accepted (e.g. Pietsch, 2014) that the ageing of a painting and the change of its colours depends on many different factors, including ratios of different pigments, the ratio of pigments and binding agents, the kind and composition of the binding agent (e.g. egg, oil, resin, glue, synthetic binder) and its specific preparation (e.g. boiled, siccativated, bleached) as well as the composition and thickness of the original and potential later coatings. In addition, the painting's individual history in terms of storage or display conditions, the light sensitivity of the individual components as well as earlier restorations need to be taken into account. In other words, the ageing and colour change would have to be investigated in detail for each individual painting, while general conclusions on, e.g. the colour change per century do not appear feasible.

We finally briefly discuss the possibility of providing a total error estimate of AOD values inferred from colour ratios in historic paintings. Such a total error estimate, applicable to many different paintings would of course be highly desirable. However, due to the complexity of the problem, the impacts of different parameters and processes – most of which are generally not well constrained – the determination of a total error estimate appears to be impossible in our opinion. It may be possible to establish a total error estimate for individual paintings, but not for the entirety of all paintings.

## 5 Conclusions

In this study we scrutinised the robustness of aerosol optical depth estimates from red-green colour ratios extracted from photographs of historic paintings. We emphasise that we do not challenge that colours and colour ratios in paintings of the evening sky may be affected by the stratospheric aerosol loading and volcanic effects thereon. But we question, whether quantitative information on the aerosol optical depth can be estimated. As shown by sensitivity studies using the SCIATRAN radiative transfer model, the red-green colour ratios do depend on the aerosol optical depth – as expected – but they also depend on several other factors, which are generally not known or only poorly constrained for past volcanic eruptions. We investigated the impact of the aerosol particle size distribution, total ozone column, surface albedo and the specific viewing geometry on colour ratios. The impact of surface albedo is only minor and so is the effect of the ozone column – except for large solar zenith angles. Of crucial importance is the aerosol particle size distribution, which has a strong impact on the spectral dependance of radiation scattered by the aerosols and hence on the red-green colour ratios. The effect is so large that the determination of aerosol optical depth from red-green ratios does not appear feasible in a robust way for a large range of SZAs. In addition, for SZAs exceeding 90° the red-green colour ratios – irrespective of the specific wavelengths or wavelength ranges chosen – vary strongly with SZA making a quantitative estimate of the AOD very difficult, if not impossible, particularly if the actual SZA is not known. Also, the exact solid angle for which colour information is extracted from the photographs of paintings needs to be determined and also used for the radiative transfer simulations. Otherwise, large systematic errors have to be expected for the optical depth estimates. This effect is particularly pronounced for viewing zenith angles close to 90° and SZAs exceeding 70°. It was also found to be essential, to compare the colour ratios extracted from photographs to the correct quantity simulated

with a radiative transfer model: The modelled colour ratios should be based on simulations of evening sky radiances and not on diffuse downward fluxes, as in some earlier studies. It is also questionable, whether the original colours of a painting can be considered realistic. In addition, the long-term change of the colours in a painting would have to be carefully investigated for
each individual painting and estimates of the original colours may be highly uncertain.

We finally emphasize that we do not categorically exclude the possibility of inferring AOD from paintings. It may be possible for individual paintings, for which all relevant parameters are sufficiently well constrained in order to reduce the related errors.

*Code availability.* The SCIATRAN radiative transfer model can be downloaded from the following website: https://www.iup.uni-bremen.de/sciatran/ (last access: October 4, 2022).

*Author contributions.* CvS designed the study, carried out the SCIATRAN simulations with assistance by AR and AL and wrote an initial version of the manuscript. All authors discussed and edited the manuscript. AH contributed an initial version of section 4.1.

*Competing interests.* The authors declare that they have no competing interests.

*Acknowledgements.* This work was supported by the Deutsche Forschungsgemeinschaft (project VolARC of the DFG research unit VolImpact FOR 2820, grant no. 398006378). We are indebted to the Institute of Environmental Physics of the University of Bremen – particularly to Dr.
Vladimir Rozanov and Prof. Dr. John P. Burrows FRS – for providing the SCIATRAN radiative transfer model. CvS thanks Renate Kühnen and Kilian Heck (both Greifswald) for very helpful discussions.

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
