# Peer review of "Is it possible to estimate aerosol optical depth from historic colour paintings?"

_Climate of the Past, 2022_

## Author Comment (AC1)

**Reply to comments by Anders Svensson:**

> *Note that our responses are indented, bold and italicized*

The manuscript is concerned with estimating the uncertainties related to deriving the Aerosol Optical Depth (AOD) from the coloring of the sky in paintings made at the time of large volcanic eruptions. As such, the manuscript provides a critical comment to existing publications that are aiming at deriving such values. Using an atmospheric radiative transfer model, the manuscript presents a number of sensitivity studies by varying a number of parameters that may lead to uncertainties in the color-based AOD estimates. The manuscript is well written and easy to follow and the various uncertainties brought to the table appear relevant.

> ***Reply:  We thank the reviewer for his positive and constructive review.***

I do not have comments on the radiative transfer model for which I am not an expert, but I have two suggestions for the authors to consider:

My first point concerns the 'true' error related to AOD estimates from paintings. For historical volcanic eruptions, there are ice-core based estimates of the sulfate deposition in both Greenland and Antarctica that provide an independent estimate of the stratospheric sulfate aerosol loading, which in turn can be translated into an stratospheric AOD (Gao et al., 2007;Gao et al., 2008;Sigl et al., 2015). Can those estimates be applied to give an independent estimate of the accuracy of the paintings derived AOD? Of course, the ice-core estimates can be questioned themselves, but a reasonable agreement between the two independent approaches would nevertheless suggest that both methods are providing AOD estimates that are in the right order of magnitude, at least. Likewise, a large disagreement between the two methods would suggest that at least one of them has very large uncertainties. Maybe this comparison has already been done in another study? It seems like a quite obvious comparison to make?

> ***Reply: This is a very good idea and a comparison with independent data sets (including AOD estimates based on ice cores) has actually already been made by Zerefos et al. (2014) (see their Figures 4 and 5). This comparison shows significant differences between the AOD values estimated from the different techniques (including analysis of ice cores and of historic transmission measurements). The agreement between the datasets is reasonable for the major eruptions with AOD values exceeding 0.1. But for the weaker eruptions the AODs estimated from the paintings are systematically larger than the ones from the other data sets. Figure 5 in Zerefos et al. (2014) shows comparisons of 50-year mean AODs from the different data sets and the AODs estimated from the paintings are systematically about 1 order of magnitude larger than the other data sets.***
>
> ***Zerefos et al. (2014) also list in Appendix D their AOD values together with values from different other studies. In order to investigate the differences, we determined averaged AOD values considering only the years for which data from the two datasets to be compared are available. We obtained the following results:***
>
> ***Zerefos vs. Robertson:***
> $AOD_{Zerefos}$ ***= 0.174 $\pm$ 0.107 (Mean value and standard deviation)***
> $AOD_{Robertson}$ ***= 0.017 $\pm$ 0.038***
>
> ***Zerefos vs. Crowley & Unterman:***
> $AOD_{Zerefos}$ ***= 0.198 $\pm$ 0.121***

$AOD_{Crowley} = 0.031 \pm 0.058$

**Zerefos vs. Sato:**

$AOD_{Zerefos}\ = 0.180 \pm 0.093$

$AOD_{Sato}\quad = 0.0175 \pm 0.022$

**Zerefos vs. Stothers:**

$AOD_{Zerefos}\ = 0.226 \pm 0.124$

$AOD_{Stothers} = 0.032 \pm 0.044$

*Figure 1 below depicts all the datasets listed in Appendix D of Zerefos et al. (2014). Apparently, the mean AOD values do differ by about an order of magnitude. As the Figure below demonstrates, the differences reach two orders of magnitude in some cases. It is also obvious that the AOD values estimated from the historic paintings are almost all larger than 0.1. Significantly smaller values are not inferred.*

*A brief discussion on the differences between the different AOD data sets compared in Zerefos et al. (2014) was included in section 4 of the revised version of our manuscript.*

[Figure]

*Figure 1: Comparison of AOD values retrieved from historic paintings (in red) with independent estimates, as described in our reponse. All data points were taken from the table in the appendix of Zerefos et al. (2014). The red dashed lines corresponds to AOD = 0.3.*

*References:*

*Crowley, T. J. and Unterman, M. B.: Technical details concerning development of a 1200 yr proxy index for global volcanism, Earth Syst. Sci. Data, 5, 187–197, doi:10.5194/essd-5-187-2013, 2013.*

*Robertson, A., Overpeck, J., Rind, D., Mosley-Thompson, E., Zielinski, G., Lean, J., Koch, D., Penner, J., Tegen, I., and Healy, R.: Hypothesized climate forcing time series for the last 500 years, J. Geophys. Res., 106, 14783–14803, 2001.*

*Sato, M., Hansen, J. E., McCormick, M. P., and Pollack, J. B.: Stratospheric aerosol optical depths 1850–1990, J. Geophys. Res., 98, 22987–22994, 1993.*

*Stothers, R. B.: Major optical depth perturbations to the stratosphere from volcanic eruptions: pyrheliometric period 1881–1960, J. Geophys. Res., 101, 3901–3920, 1996.*

*Stothers, R. B.: Major optical depth perturbations to the stratosphere from volcanic eruptions: Stellar extinction period 1961–1978, J. Geophys. Res., 106, 2993–3003, 2001.*

My second point concerns an overall error estimate for the historic color painting method for estimating the stratospheric aerosol optical thickness based on the uncertainties introduced in the present manuscript. In the manuscript, we are provided with numerous figures showing the AOD sensitivity to factors such as particle size distribution, wavelength, solar zenith angle, albedo, azimuth angle, etc. All of those dependencies certainly leave the impression 'that the uncertainties of the estimated aerosol optical depths are so large that the values have to be considered highly questionable', as mentioned in the abstract. However, how large are the uncertainties 'typically' in a real-case scenario? If we add up all of the uncertainties using a realistic range of values for the parameters discussed in the study, do we then end up with 5% or 50% uncertainty on the final result? If the total uncertainty is in the range below say 50%, the method may still be applicable, eg if there are several paintings of the (sky of the) same eruption that may provide independent evidence. If the final uncertainty estimate is large however, say above 50%, the entire approach of using paintings for estimating the AOD becomes questionable. Therefore, some kind of summary providing a combined uncertainty from all of the discussed parameters would be quite helpful. Also, an estimate of the relative uncertainty contribution from each investigated parameter would be helpful again using a realistic range of parameters. If possible, some uncertainty estimates/ranges could be provided in a table? This may provide some useful guidance for future studies of what knowledge is needed to make constrained AOD estimates from paintings. Maybe, in some cases, there is independent evidence of say the position of the Sun or the time of the day when the picture was painted? Likewise, we may become wiser in the future about what to expect from the particle size distribution related to large volcanic eruptions. Thus - wearing an optimistic hat - it could be that some of the uncertainty ranges discussed in the manuscript could be significantly reduced or even eliminated for specific paintings/eruptions?

> *Reply: Another good idea, which was also suggested by reviewer #3. We thought about a total error estimate already when writing the manuscript and decided to omit it, because we believe it is not possible to reduce the complexity of the problem and the impacts of the different effects and parameter uncertainties in a single number. Also, the large differences between the AODs in Zerefos et al. (2014) and the independent data sets discussed above question the possibility of a total error estimate. We do think, however, that it is possible to carry out a total error estimate for individual paintings, if some of the critical parameters are well known or can be sufficiently constrained. Also following the comments by reviewer #3 we adjusted the basic message of the paper and we now state that it may be possible to infer quantitative information on the AOD from individual paintings, whose history is well known and for which all relevant parameters are known sufficiently accurately.*

> *We understand that it would be desirable to have a total error estimate or at least a paragraph describing the issues related to provide one. We added a discussion to the results section, providing a justification for our assumption that a total error estimate is not possible. The following aspects are important here.*

*A major problem – for paintings showing scenes with the sun below the horizon – is the strong SZA-dependence of the red-green colour ratio (see Fig. 3 of the revised version of the paper). As already pointed out in the paper, without exact knowledge of the SZA, an estimation of the AOD is not possible. Even if the painting is finished in, e.g. 30 minutes during the sunset, the SZA (and with it the colours) will have changed significantly.*

*Regarding the problems discussed in section 4 (How realistic were the colours on the day the painting was finished & how did the colours change over time), we strongly believe that general estimates of these effects are not possible and should not be attempted, in order to avoid implying a false sense of reliability of the results. Again, for an individual painting robust error estimates might be possible, but not for the entirety of all paintings.*

*The effects of uncertainties in the assumptions of the amount (and profile) of ozone or the surface albedo are rather small and not the main issues.*

References:

Gao, C. C., Robock, A., and Ammann, C.: Volcanic forcing of climate over the past 1500 years: An improved ice core-based index for climate models, Journal of Geophysical Research-Atmospheres, 113, D23111, 10.1029/2008jd010239, 2008.

Gao, C. H., Oman, L., Robock, A., and Stenchikov, G. L.: Atmospheric volcanic loading derived from bipolar ice cores: Accounting for the spatial distribution of volcanic deposition, Journal of Geophysical Research-Atmospheres, 112, 10.1029/2006jd007461, 2007.

Sigl, M., Winstrup, M., McConnell, J. R., Welten, K. C., Plunkett, G., Ludlow, F., Büntgen, U., Caffee, M., Chellman, N., Dahl-Jensen, D., Fischer, H., Kipfstuhl, S., Kostick, C., Maselli, O. J., Mekhaldi, F., Mulvaney, R., Muscheler, R., Pasteris, D. R., Pilcher, J. R., Salzer, M., Schüpbach, S., Steffensen, J. P., Vinther, B. M., and Woodruff, T. E.: Timing and climate forcing of volcanic eruptions for the past 2,500 years, Nature, 523, 543-549, 10.1038/nature14565, 2015.

---

## Author Comment (AC2)

**Reply to comments by anonymous reviewer #2:**

> *Note that our responses are indented, bold and italicized*

The manuscript challenges the notion that historical paintings can be used to quantitively assess the amount of stratospheric sulphate aerosols following major volcanic eruptions, based in particular on a series of sensitivity tests with an atmospheric radiative transfer model.

In general the manuscript is clear and well written, and it could be of interest to Climate of the Past readers.

I think that already in the introduction the authors should clarify the premises of this study, as the readers may not be familiar with all the background facts, for instance by concisely but explicitly addressing the following issues:

Why do you focus on near-horizon radiance (e.g. evenings)?

Why do you focus on the red/green ratio?

How do you know that the painting is depicting the evening?

How do you know that the painter's style was realistic in reproducing the colors?

How do you know that pigment conservation allows for estimating the original colors faithfully?

> *Reply: We thank the reviewer for these constructive comments. We tried to incorporate the reviewer's suggestions in the introduction and hope they enable the reader to better understand the context of the study. We are, however, not entirely certain we fully understand the reviewer's comments, because we are criticizing some of these assumptions, i.e. the last three questions posed by the reviewer. We now at least mention these aspects in the introduction and we hope that by doing so we did follow the reviewer's intention.*

Specific comments

9) Krakatoa?

> *Reply: Thank you, changed!*

55) "The troposphere was assumed to be free of aerosols". This appears to be a strong assumption. Was there a basic sensitivity test at least, to justify this? No reference is provided either, of why this assumption should hold.

> *Reply: This is a good point, and we carried out a simulation with tropospheric aerosols. Adding a tropospheric component with an AOD of 10% of the total AOD (i.e. 0.03 for a total AOD of 0.3) leads to only very small differences (less than 0.5 %) in the colour ratios for SZA < 90 deg. For SZA > 90 deg, the differences can reach up to 10%, but the overall conclusions of the study are not affected. For most (or all) scenarios associated with historic paintings, the tropospheric AOD and the characteristics of the aerosol are not well known and add further complications when trying to infer AOD information from the paintings. We added a statement to the paper briefly discussing tropospheric aerosols.*

Table 1) If I interpreted the parameter values correctly, the central case is that of painters reproducing on canvas what they see in front of them, just before sunset, while giving their back to the setting sun. Is that correct? Maybe it's worth spelling this out.

*Reply: Thanks for this question. The basic viewing geometry is that the observer is looking in the solar direction, not in the anti-solar direction. We added the following sentence to section 2.1 to make this point clear:*

> *"Note that the range of solar azimuth angles considered implies that the observer is looking in sun-ward direction."*

*In addition, the following statement was added to the sentence in the caption of Table 1, where the meaning of the solar azimuth angle is explained:*

> *"i.e. an SAA of 0° means that the observer is looking in the direction of the sun;"*

59) Since the red/green ratio is relevant here, and the application of tristimulus values implies using specific wavebands, it would be worth seeing a curve of RI vs wavelength

*Reply: We are not sure we understand this comment properly. You are asking about the wavelength dependence of the refractive index, right? This wavelength dependence is considered in this study. However, the spectral dependence of the real part of the refractive index is very small. It changes by only 0.003 when going from 500 nm to 700 nm. The refractive index data are taken from the OPAC (Optical Properties of Aerosol and Clouds) database implemented in SCIATRAN.*

*We now mention in the paper that the spectral dependence of the real part of the refractive index of sulfate aerosols is considered in the simulations.*

85) Why not showing a plot of the tristimulus values wavelength dependence?

*Reply: We did not include it at first, because in a previous publication (where we showed this plot) one reviewer asked to remove it, because it is standard textbook knowledge. But we follow the reviewer's suggestion here and have now included such a plot (Figure 2 of the revised manuscript).*

---

## Author Comment (AC3)

**Reply to comments by anonymous reviewer #3**

*Note that our responses are indented, bold and italicized*

This study raises a number of important uncertainties that were not considered in the Zerefos et al., 2007 and 2014 methodology, in particular regarding the particle size distribution and stratospheric ozone concentration. It acts as an interesting and useful continuation of research on the question of whether historical paintings encode quantitative environmental information. However, I have four major comments that question the conclusion that "quantitative determination of the AOD from colours in paintings is generally not possible" — while agreeing that error quantification of the Zerefos methodology is an important next step.

*Reply: We thank the reviewer for his/her insightful and constructive comments and we have taken most of them into account when revising the manuscript.*

*We agree with the reviewer that the overall conclusion could have been phrased a bit more carefully and precisely. We think that it is probably feasible to infer quantitative information on the AOD from individual paintings, for which all relevant boundary conditions are well known (ozone, particle size etc.) and for which potential temporal changes in the colours have been quantified with appropriate techniques. However, we still think that a reliable determination of the AOD for many paintings and with crude assumptions is not possible. This assessment is also based on the comparison of the derived AOD values with independent data sets, as presented in Zerefos et al. (2014) (please also see our response to the comment by reviewer 1). On average, the AOD values determined based on the analysis of paintings are about an order of magnitude larger than the values from the independent data sets.*

*We added a paragraph to the discussion section to discuss our modified understanding of the applicability of the method.*

**Major comments**

**1)** More justification for the choice of parameters used in the sensitivity studies should be provided. Having a sense of how these perturbations compare to values in a typical volcanic eruption would help contextualize the magnitude of the resulting errors.

*Reply: This is a good idea and we added text to the corresponding sections of the manuscript (see also reponses below).*

As one example, regarding the claim that the particle size distribution is a strong confounding variable, it would be useful to provide more justification for the choice of 350 and 450 nm as perturbed parameters. A study is cited of observed values > 400 nm after Pinatubo. The authors do mention a number of other uncertainties regarding the particle size distribution and this sensitivity test, e.g., whether the distribution is log-normal, what are reasonable median values, whether the median scales with the strength of the volcanic eruption, how the distribution depends on the observation geometrics. Another issue appears to be how the median value varies as a function of altitude and latitude (cf. Figure 1b in Bingen et al., 2004). Given these various uncertainties, it is not clear to me whether median particle size can be considered a systematic source of error.

*Reply: The reviewer raises an interesting point and perhaps our explanations were not fully clear (although we believe that all relevant statements were included in the section). The main issue here is that the variability of the size distribution of stratospheric sulfate aerosols after volcanic eruptions is not well understood, but is has a significant effect on the colours*

*of the evening sky. Until recently, many (if not most) members of the scientific community thought that the size of stratospheric sulfate aerosols increases after volcanic eruptions, mainly because this is well documented in in-situ and also satellite measurements for the Pinatubo eruption. Only recently it became clear that for many (probably mainly smaller) eruptions, the aerosol particles may become smaller on average (Thomason et al., 2021). The exact reasons are not fully understood and the topic of current research. Most likely, larger volcanic eruptions (Pinatubo and stronger) will lead to larger particles and this is also what we assumed. We are really sorry, but the current level of understanding does not allow making any more plausible assumptions than the ones we have made. We modified the text and hope that the motivation (and limitations) of the chosen parameters are now more transparent.*

*The dependence of the size parameters on altitude and latitude mentioned by the reviewer, are an additional issue, which complicates things even further.*

*Regarding the term „systematic error" in the last sentence of the reviewer comment above: We agree that from the perspective of a large number of paintings (and inferred AOD values), the unknown aerosol particle size distribution may not be a systematic error in the usual sense. However, the assumption that the unknown size distribution can be considered a random error that cancels out if a sufficiently large number of paintings is analyzed is not justified either, in our opinion. In addition, despite new results on smaller eruptions the scientific community still expects a general increase of particle sizes after very strong eruptions. As only strong eruptions are well known from the past and only they can be attributed to paintings it is reasonable to expect a "systematic" issue related to larger aerosol particles.*

*Furthermore, our intention was to express that for the determination of AOD values from individual paintings the missing knowledge on the aerosol particle size distribution constitutes a systematic error. For a larger ensemble of paintings this may not be the case. Please also note that we do not use the term „systematic error" in the context of the particle size distribution. But thank you for pointing this out!*

A related specific issue concerns that Pinatubo was a large eruption (VEI 6), and similarly VEI 6 eruptions (between years 1500-1900) in the Zerefos study have AODs around 0.4-0.6, which is outside the range of AODs shown in Fig. 2 (going to 0.3). It would be useful to include the curves for an AOD value of 0.5 or 0.6 to show how the uncertainty introduced by particle size distribution compares with the full range spanned by AOD values considered in Zerefos et al., 2007 (e.g., their Fig 6).

*Reply: We are sorry, but we do not fully agree here. Zerefos et al. show AOD values of up to about 0.6, but they are not found in the independent data sets discussed in Zerefos et al. (2014). The maximum AOD values in the independent data set are about 0.3 and this was also the reason, why we chose 0.3 as the upper value. As mentioned above, there is on average about one order of magnitude difference between the AOD values inferred from the paintings and the ones based on independent methods (see also Figure 1 below).*

[Figure]

*Figure 1: Comparison of AOD values retrieved from historic paintings (in red) with independent estimates, as described in our reponse. All data points were taken from the table in the appendix of Zerefos et al. (2014). The red dashes line corresponds to AOD = 0.3.*

Similarly, justification for the choice of ozone parameters would also be helpful. It seems that the influence of volcanic aerosols on stratospheric ozone is not yet fully understood. The different perturbations in this study could, moreover, co-vary, which has not been considered. For instance, it could, conceivably, be that a larger volcanic eruption does indeed increase the median particle size, but decreases the stratospheric ozone concentration, such that these two errors cancel to some extent. Some research indicates that stratospheric ozone could decrease if there is a natural (biogenic) source of bromine or hydrogen halides (e.g., Klobas et al., 2017, Ming et al., 2020). The large uncertainties, both in the individual processes and possible interactions/compensations among different factors also introduce substantial uncertainty in the sensitivity tests shown here that is not yet discussed.

*Reply: We agree with the reviewer that a more detailed discussion on the variability of the total ozone column (TOC) and the choice of TOC values is useful. We added a paragraph on this to section 3.4, following the reviewer's suggestions. We agree that some of the effects discussed may partly cancel each other, but it is not possible to address all possible combinations of effects here. The effect of volcanic aerosols on the ozone layer is of course quite important and interesting and the following aspects are now also briefly discussed in section 3.4:*

*The effect of the eruptions of El Chichon and Mount Pinatubo on the globally averaged TOC was on the order of a few percent and is hence smaller than the typical seasonal variation in TOC.*

*Reductions in TOC due to volcanic eruptions are only expected in periods with anthropogenically enhanced stratospheric halogen loading. In the past centuries (and also in the future), volcanic eruptions are rather expected to lead to an increase in TOC (e.g. Tie and Brasseur, 1995).*

*Reference:*

*Tie, X., and G. Brasseur (1995), The response of stratospheric ozone to volcanic eruptions: Sensitivity to atmospheric chlorine loading, Geophys. Res. Lett., 22, 3035–3039, doi:10.1029/95GL03057.*

**2)** As a continuation of the points raised in 1), it would be useful to have a synthesized error estimate – that is, jointly considering the influence of the various sensitivity studies. Such error bounds would allow for a like-for-like comparison with the Zerefos results (e.g., as a function of AOD and zenith angle, as in Zerefos et al., 2007 Tables 2 and 3). Such an overall error estimate would also give a sense of how the uncertainties involved in this methodology compare with uncertainties present in other methodologies to estimate historical volcanic AODs, such as from ice cores.

That said, an overall error bound seems difficult to estimate, given a lack of knowledge of how these error sources combine: additively? some compensation as discussed in 1)? It is not clear whether the error sources would combine to yield *iid* observational error, adding noise to the estimates of the paintings. In that case, that a signal in the R/G ratios can nevertheless be extracted, despite the noise, is surprising and would point to a strength, rather than a weakness, in the methodology.

In conclusion, it seems that overall error bounds are necessary before one can make the argument that a quantitative estimate of volcanic AOD from historical paintings is not possible.

> *Reply: This is in principle a very good idea and reviewer #1 made a similar suggestion. We actually thought about a total error estimate already when writing the manuscript and decided to omit it, because we believe it is not possible to reduce the complexity of the problem and the impacts of the different effects and parameter uncertainties to a single number. We do think, however, that it is possible to carry out a total error estimate for individual paintings, if all of the critical parameters are well known or can be sufficiently constrained. But an overall estimate of this technique to be applied to any painting or a large number of paintings is in our view not possible. Also, the large differences between the AODs in Zerefos et al. (2014) and the independent data sets discussed above question the possibility of a total error estimate.*
>
> *But we understand that it would be desirable to have such a total error estimate or at least a paragraph describing the issues related to provide one. We added a discussion to the results section, providing a justification for our assumption that a total error estimate is not possible. The following aspects are relevant in this context:*
>
> 1. *A major problem – for paintings showing scenes with the sun below the horizon – is the strong SZA-dependence of the red-green colour ratio (see Fig. 3 of the revised version of the paper). As already pointed out in the paper, without exact knowledge of the SZA, an estimation of the AOD is not possible. Even if the painting is finished in, e.g. 30 minutes during the sunset, the SZA (and with it the colours) will have changed significantly.*
> 2. *Regarding the problems discussed in section 4 (How realistic were the colours on the day the painting was finished & how did the colours change over time), we strongly believe that general estimates of these effects are not possible and should not be attempted, in order to avoid implying a false sense of reliability of the results. Again, for an individual painting robust error estimates may be possible, but not for the entirety of all paintings.*
> 3. *The effects of uncertainties in the assumptions of the amount (and profile) of ozone or the surface albedo are rather small and not the main issues.*

**3)** The art historical analysis in Sec. 4.1 focuses only on a single painter, Caspar David Friedrich, who was known to be more spiritual and less empirical than many of his contemporaries, such as John Constable, JMW Turner, or Johan Christian Dahl. As one example, the German thinker Johann Wolfgang von Goethe became aware of Luke Howard's cloud studies and expounded them in intellectual circles in Germany. As described in Richard Hamblyn's *The Invention of Clouds: How an Amateur Meteorologist Forged the Language of the Skies*, whereas many painters, such as Constable and Dahl actively engaged with these scientific ideas, Friedrich was an exception in categorically dismissing scientific ideas as incompatible with his artistic practice (e.g., refusing Goethe's request to provide illustrations for Goethe's 1817 essay on Howard, saying the project would 'undermine the whole foundation of landscape painting, *Hamblyn, p. 221*).

It is understood that this section is not an exhaustive art historical analysis. Yet the balance of this section needs improvement. It could note the general spirit of empiricism and greater overlap between the arts and sciences that was present in the period considered from 1500-1900 – rather than simply selecting one painter to make the point that artwork does not contain quantitative environmental information. Environment is, of course, not the only factor that influences artwork, but there is evidence that painters do depict the environment with some sort of fidelity. This evidence can be qualitative (e.g., the Turner quote at the end of Zerefos et al., 2007, Paul Cézanne's critique of Monet that he was "only an eye…But my God!... what an eye!"), but also more quantitative (e.g., Olson et al., 2003, Aragon et al., 2006, Fikke et al., 2017).

> *Reply: The authors thank the reviewer for the sophisticated and insightful comment on the relation of sciences and arts in the Romantic period. It is undoubtedly true that the interweaving of atmospheric studies in science and art in the early 19th century was more complex and multi-layered. The aim of this discussion section is therefore to emphasise the current urgency and value of transdisciplinary work between art historians and atmospheric scientists with regard to ecological issues. We adjusted the text and added additional text in order to address the points raised by the reviewer. Additions to the text also intend to reinforce the need of a greater consideration of different art historical examples for an extensive transdisciplinary study. Moreover, we now emphasize more strongly that this part of the discussion shall explicitly be understood as a commentary on the previous scientific investigation and focusses on C. D. Friedrich's painting "Greifswald im Mondschein". In our view, Friedrich's painting is not an unfitting but a particularly exciting example, as the painter dealt with the interface between ecological awareness, philosophical reflections on the subjectivity of perception and questions of immanence and transcendence in human's encounter with nature in a rather complex way.*

**4)** My fourth major comment relates to the 'tone' of the manuscript, which gives the impression of dismissing any quantitative skill of the Zerefos et al. methodology (e.g., that the Zerefos results "have to be considered highly questionable"). The uncertainties mentioned in points 1) and 2), however, imply that the ultimate magnitude of the error is far from certain. It therefore appears somewhat misleading and unfortunate to entirely disregard this methodology, especially given the paucity of records of historical volcanic AOD.

Perhaps the total errors will someday be shown to be too large such that the Zerefos methodology does not have any quantitative skill, but it is premature to make this conclusion given uncertainties in 1) the perturbed parameters in the sensitivity studies and 2) potential compensations among the different factors considered here. Instead, the authors could consider changing the language to say that uncertainties considered are avenues for further research. Reducing uncertainty about these important issues would allow for better constraining volcanic AOD from historical images.

*Reply: As already mentioned above, we changed the „tone" of the manuscript and now do not categorically exclude the possibility that AOD values can be inferred from individual paintings. We did change several passages of the paper and now clearly state that there may be quantitative information on aerosol optical depth in individual paintings, if all relevant parameters are well known and potential long-term changes in the colours have been quantified in a robust way.*

*Our initial judgement was also based on the large differences between the AODs determined by Zerefos et al. (2014) and the AODs obtained from other sources. We still believe that values that are on average an order of magnitude off are „highly questionable" (we did weaken the sentence in the abstract, though and adjusted several statements that included the word „impossible"). We did not address these differences in the initial version of our manuscript, but they are now briefly discussed.*

**Minor comments**

**1)** The authors might consider including a schematic illustrating the key radiative processes and quantities, e.g., diffuse downward radiation vs. near-horizon radiance, solar zenith angle, to increase readability for a broader audience. In such a schematic, it could also be worthwhile to show a sunset with bluer sky towards the zenith and increasingly red colors towards the horizon to illustrate the issue of irradiance vs. near-horizon radiance.

> *Reply: This is a good idea and we included such a schematic in the revised version of the manuscript (now Fig. 1 in the revised version of the manuscript)*

**2)** Zerefos et al., 2007 say that there are large errors when the zenith angle exceeds 90 degrees: "In our calculations we used the diffuse irradiance over the whole hemisphere at the given wavelengths. However if we calculate the R/G ratios shown in Figs. 1 and 2 using the integral of the calculated radiances within 20 degree azimuth, around the setting sun and for zenith angles 70–90deg, then we would be able to simulate with the model the paintings' ratios even better. We note here that the radiative transfer solver included in libRadtran is only a pseudo-spherical and not a fully spherical code, and therefore its accuracy for radiance calculations is limited at high SZAs.". The current manuscript says, "One issue is that they did not calculate the radiance ratio for a near-horizon viewing geometry, but calculated the ratio of the diffuse downward fluxes at the two wavelengths, although the near-horizon radiance would be the correct quantity in this context." I found this discrepancy hard to understand. Could the authors please clarify why Zerefos et al., 2007 wrote that the accuracy of radiance calculations is limited at high solar zenith angles, but it is no longer a problem – have the models improved since 2007 to eliminate this issue? They also seem to address the concern of diffuse downward fluxes vs. near-horizon radiance in their manuscript and find that it little influences results (as cited above), so I was surprised to see this difference so prominently highlighted.

> *Reply: The reviewer addresses different points and we will respond to them one-by-one.*
>
> 1. *Yes, Zerefos et al. mention the difference diffuse irradiance vs. near-horizon radiance explicitly in the paper, but they do not discuss that this is actually a major issue. In our opinion this is a problem, because these two quantities exhibit different dependencies on AOD and SZA, as shown in our manuscript (Figure 8 of the revised version). The near-horizon radiance would be the correct quantity to use, but because they did not trust the libRadtran simulations, Zerefos et al. use the diffuse irradiance (or diffuse downward flux) instead. From a radiative transfer point-of-view this is not justified and will lead to large errors in the estimated AOD (see left panel of Figure 8 in our revised manuscript).*

*Note that one of the reviewers of the Zerefos et al. (2007) paper also already criticized the use of the diffuse irradiance instead of the near-horizon radiance.*

2. *With SCIATRAN we have simulated the radiances/diffuse fluxes with a better representation of the spherical geometry of the atmosphere, as described in the paper. Single scattering is calculated in fully spherical geometry and an approximation is employed for the multiple scattering contribution. In another recent paper (Lange et al., 2022) we showed that this setup is a good approximation with errors of only a few % (compared to a full and time-consuming treatment of the multiple scattering contribution) for the visible spectral range and a near horizon viewing geometry (see Fig. 12b in Lange et al. (2022)).*

3. *The ratio between the diffuse downward flux and the near-horizon radiance does depend on the SZA and also on the AOD (Figure 8 of our revised paper). However, in Zerefos et al. a constant factor is employed to scale the irradiance simulations to better agree with the red-to-green ratios extracted from the paintings. This scaling factor is (in part) required to compensate for the fact that the diffuse irradiance is simulated rather than the near-horizon radiance. The value of this scaling factor is, however, not given in Zerefos et al. (2007).*

*Reference:*

*Lange, A., Baumgarten, G., Rozanov, A., and von Savigny, C.: On the colour of noctilucent clouds, Ann. Geophys., 40, 407–419, doi.org/10.5194/angeo-40-407-2022, 2022.*

**3)** Typo 'Krakatoa' line 9 in abstract

*Reply: Thank you, corrected!*

**4)** Lines around 85, a reference could perhaps be given for the CIE colour matching functions?

*Reply: Following the comments by reviewer #2 we added a plot with the CIE colour matching functions (New Fig. 1) together with the corresponding references. We also added a reference to this new Figure in the paragraph pointed out by the reviewer #3.*

**References:**

Aragón, J.L., Naumis, G.G., Bai, M., Torres, M. and Maini, P.K., 2008. Turbulent luminance in impassioned van Gogh paintings. Journal of Mathematical Imaging and Vision, 30(3), pp.275-283.

Bingen, C., Fussen, D. and Vanhellemont, F., 2004. A global climatology of stratospheric aerosol size distribution parameters derived from SAGE II data over the period 1984–2000: 1. Methodology and climatological observations. Journal of Geophysical Research: Atmospheres, 109(D6).

Fikke, S.M., Kristjánsson, J.E. and Nordli, Ø., 2017. Screaming clouds. Weather, 72(5), pp.115-121.

Hamblyn, R., 2002. The invention of clouds: How an amateur meteorologist forged the language of the skies. Pan Macmillan.

Klobas, E. J., Wilmouth, D.M., Weisenstein, D.K., Anderson, J.G. and Salawitch, R.J., 2017. Ozone depletion following future volcanic eruptions. Geophysical Research Letters, 44(14), pp.7490-7499.

Ming, A., Winton, V.H.L., Keeble, J., Abraham, N.L., Dalvi, M.C., Griffiths, P., Caillon, N., Jones, A.E., Mulvaney, R., Savarino, J. and Frey, M.M., 2020. Stratospheric ozone changes from explosive tropical volcanoes: Modeling and ice core constraints. Journal of Geophysical Research: Atmospheres, 125(11), p.e2019JD032290.

Olson, D.W., Doescher, R.L. and Olson, M.S., 2003. Dating Van Gogh's" Moonrise".

---

## Author Response (AR2)

**Reply to comment by reviewer #2:**

Dear Dr. Rousseau,

we considered the comment by reviewer #2 by mentioning that the average tropospheric aerosol optical depth in modern times is larger than the value assumed in our paper, as suggested by the reviewer. We also cite the reference mentioned by reviewer #2.

Many thanks for your help with our manuscript!

Sincerely,

Christian von Savigny